# A Single Amino Acid Substitution in RFC4 Leads to Endoduplication and Compromised Resistance to DNA Damage in *Arabidopsis thaliana*

**DOI:** 10.3390/genes13061037

**Published:** 2022-06-09

**Authors:** Kan Cui, Lei Qin, Xianyu Tang, Jieying Nong, Jin Chen, Nan Wu, Xin Gong, Lixiong Yi, Chenghuizi Yang, Shitou Xia

**Affiliations:** 1Hunan Provincial Key Laboratory of Phytohormones and Growth Development, College of Bioscience and Biotechnology, Hunan Agricultural University, Changsha 410128, China; ck0601@stu.hunau.edu.cn (K.C.); leiqin@stu.hunau.edu.cn (L.Q.); chuxuantingnasha@stu.hunau.edu.cn (X.T.); njy@stu.hunau.edu.cn (J.N.); wunan@stu.hunau.edu.cn (N.W.); xingong@stu.hunau.edu.cn (X.G.); ychz1995@stu.hunau.edu.cn (C.Y.); 2Hunan Academy of Agricultural Sciences, Changsha 410125, China; chenjin_0424@126.com (J.C.); xiong-mao.168@163.com (L.Y.); 3Changsha Technology Innovation Center for Phytoremediation of Heavy Metal Contaminated Soil, Hunan Academy of Agricultural Sciences, Changsha 410125, China

**Keywords:** *Arabidopsis thaliana*, replication factor C, resistance, endoduplication, DNA damage repair

## Abstract

Replication factor C (RFC) is a heteropentameric ATPase associated with the diverse cellular activities (AAA+ATPase) protein complex, which is composed of one large subunit, known as RFC1, and four small subunits, RFC2/3/4/5. Among them, RFC1 and RFC3 were previously reported to mediate genomic stability and resistance to pathogens in *Arabidopsis*. Here, we generated a viable *rfc4e* (*rfc4−1/RFC4^G54E^*) mutant with a single amino acid substitution by site-directed mutagenesis. Three of six positive T2 mutants with the same amino acid substitution, but different insertion loci, were sequenced to identify homozygotes, and the three homozygote mutants showed dwarfism, early flowering, and a partially sterile phenotype. RNA sequencing revealed that genes related to DNA repair and replication were highly upregulated. Moreover, the frequency of DNA lesions was found to be increased in *rfc4e* mutants. Consistent with this, the *rfc4e* mutants were very sensitive to DSB-inducing genotoxic agents. In addition, the G54E amino acid substitution in *At*RFC4 delayed cell cycle progression and led to endoduplication. Overall, our study provides evidence supporting the notion that RFC4 plays an important role in resistance to genotoxicity and cell proliferation by regulating DNA damage repair in *Arabidopsis thaliana*.

## 1. Introduction

The replication factor C (RFC) complex, which is an AAA+ATPase composed of one large subunit, known as RFC1, and four small subunits, RFC2/3/4/5, was first purified from the Hela cell extract of human cervical cancer and is essential for simian vacuolating virus 40 (SV40) DNA replication in vitro [1,2,3,4]. RFC subunits possess a cluster of conserved motifs that have been termed RFC boxes [5]. The four small subunits, RFC2−5, and contain seven conserved RFC boxes, II−VIII, which mainly exist in the N-terminal region. RFC box III contains the most conserved motif: a phosphate-binding loop (P-loop or Walker A domain, GxxGxGK [S/T]) that is essential for the structure and function of RFC [6,7,8]. While the conserved N-terminal structure is related to ATP binding/hydrolysis as well as binding to DNA [4], the deletion mutations of RFC1/2/3/4/5 in humans [9,10], *Arabidopsis,* and rice [11] have shown that the C-terminal region is indispensable for RFC complex formation. RFC subunits assemble in a circular arrangement via tight interactions between their C-terminal domains; therefore, due to the indispensability of each subunit, the T−DNA insertion mutants of *AtRFC2*−*5* are all lethal in *Arabidopsis* [11,12].

Moreover, all five subunits are essential for maintaining the stability of the genome in eukaryotes. The *Scrfc1* (D513N) mutant exhibited a delay in the cell cycle, increased sensitivity to DNA-damaging agents and elongated telomeres in *Sacharomyces cerevisiae* [13]. Meanwhile, both *Scrfc2−1 (*L304P*)* and *Sc**rfc2−K71R* have defects in the S-phase checkpoint, indicating that *ScRFC2* is important in both DNA replication and cell cycle checkpoints [8,14]. Temperature-sensitive *Scrfc5−1* (G43E) is sensitive to DNA-damaging agents, which can be suppressed by the overexpression of *Rad53*, the essential protein kinase responsible for DNA damage, suggesting that RFC5 is part of a mechanism transducing the DNA damage signal and slowing the S-phase progression in response to DNA damage [15,16]. The *Sprfc3−1* (R216W) mutant showed aberrant mitosis with fragmented or unevenly separated chromosomes at a restricted temperature, as well as increased sensitivity to DNA-damaging agents and UV radiation [17]. In addition, the mutation of *RFC4* in *Drosophila* leads to the premature termination of protein translation, which causes striking defects in mitotic chromosome cohesion and condensation [18].

In plants, genomic instability in RFC mutants may inhibit plant growth, trigger the DNA damage response (DDR), and lead to greatly increased frequencies of somatic recombination and heritable mutations, threatening cell survival and potentially leading to the transmission of deleterious mutations to the next generation [19,20,21]. The DDR senses genomic damage and activates cell cycle checkpoints to promote DNA repair, and double-strand breaks (DSBs) in the DNA double helix are considered to be one of the major forms of DNA damage [22]. Endoduplication is also one of the common responses to DNA damage in addition to cell cycle arrest and cell death. A lack of chromatin assembly factor 1 (CAF1) activity-induced DSB accelerates endoduplication in seedlings and leaves [23]. Zeocin treatment in wild-type *Arabidopsis*, which is known to produce DSBs, also shows increased DNA ploidy [24].

*Replication factor C subunit 1* (*RFC1*) in *Arabidopsis thaliana* was reported to play an important role in meiotic recombination and crossover formation, and in DNA double-strand break repair during meiosis [25,26,27]. The *Atrfc3−1* (G84A) mutant exhibited hyper-sensitivity to salicylic acid and enhanced resistance to virulent oomycete *Hyaloperonospora arabidopsidis* (*H. a*.) Noco2, suggesting that *AtRFC3* negatively regulates systemic acquired resistance and has crucial functions in cell proliferation and DNA replication [28,29]. However, the molecular mechanisms of the other three subunits are still unclear due to the lack of viable mutants. In this study, to reveal the functional details of *AtRFC4* in higher plants, we successfully generated viable *rfc4−1/RFC4^G54E^* (hereinafter referred to *rfc4e*) mutants by site-directed mutagenesis in the Walker A domain of RFC4. The single amino acid substitution of 54th Gly to Glu in *At*RFC4 causes developmental defects and earlier flowering. *rfc4e* mutants are also sensitive to DNA-damaging agents, leading to cell cycle arrest and endoduplication, indicating that *RFC4* is an important mediator of DNA damage repair in plant growth and resistance to abiotic stresses in *Arabidopsis*.

## 2. Materials and Methods

### 2.1. Plant Materials and Growth Conditions

The *AtRFC4* T*−*DNA line (Salk_049715) was acquired from the Arabidopsis Biological Resource Center. Plants were grown on ½ MS medium with 1% sucrose and 0.35% phytagel. Seeds were surface-sterilized for 2 min in bleach solution (added with 15% NaClO and 0.1% Tween 20), and then rinsed three times with sterilized water. The seeds were cold-treated for 3 days at 4 °C in the dark, and transferred to a growth chamber with long-day conditions of 16 h light and 8 h dark, at approximately 22 °C and 50% relative humidity, or directly planted on soil (nutrient soil: vermiculite: sand = 1:2:3); then, they were cultured in a growth room under 16 h light (22 °C)/8 h dark (22 °C) conditions.

### 2.2. Construction of Binary Vectors and Transformation

The site mutant variants of *RFC4(G54E/D)* with native promotor were generated by PCR-mediated site-directed mutagenesis, as described in a published article [30]. All primers used to generate *RFC4* site-specific mutants of *Arabidopsis* are listed in Appendix A. The variants were sequenced and, via the ClonExpress II One Step Cloning Kit (Vazyme, C112-01, China), cloned into the *pCambia 1305::3FLAG* vector and transformed into *rfc4−1* heterozygotes. *rfc4−1* plants were genotyped using the T*−*DNA left border primer, LBa1 together with the *RFC4* gene-specific primer FP2 and the *RFC4* gene-specific primer pair FP1 and FP2. For the complementation test, the full-length genomic fragment of *AtRFC4*, including its native promoter, was cloned into the *pCambia 1305::GFP* plasmid vector and transformed to *rfc4e−3* plants.

All constructs were transformed into *Agrobacterium* strain GV3101 using the electroporation method, and then transformed into plants using the floral-dip method, as described previously [31]. Seeds harvested in bulk from each plant pot were sterilized and then screened on ½ MS medium supplemented with 50 μg·mL^−1^ hygromycin. Selected homozygous mutants were used for phenotyping and transcription assays.

### 2.3. RNA Extract and RT-qPCR

For the expression analysis of genes, total RNA was isolated from whole seedlings using the Eastep™ Super Total RNA Extraction Kit (Promega, Madison, WI, USA). Reverse transcription was carried out using the GoScript™ Reverse Transcription System (Promega, Beijing, China). The RT-qPCR assay was carried out using 2 × SYBR Green Premix Pro Taq HS Premix (AG11702, Accurate Biotechnology (Hunan) Co., Ltd., Changsha, China)) and a Step-One real-time fluorescence PCR instrument (Applied Biosystems, Bedford, MA, USA). The RT-qPCR reaction system contained 10 ng cDNA, 4 µM of each primer, 5 µL 2 × SYBR Green Premix Pro Taq HS Premix, 0.2 µL ROX reference dye and 3.4 µL RNAase-free water. The RT-qPCR programming was as follows: denaturation at 95 °C for 120 s, followed by 40 amplification cycles (95 °C for 20 s, 55 °C for 20 s and 72 °C for 30 s). *AtActin1* was used as an internal housekeeping gene. Two or more independent biological replicates and three technical replicates of each sample were performed for quantitative PCR analysis. Gene-specific primers used in the experiments are listed in Appendix A.

### 2.4. Transcriptome Analysis

Total RNA was extracted by the mirVana™ miRNA Isolation Kit (Invitrogen, Waltham, MA, USA) from the 12*−*d*−*old Col*−*0 and *rfc4e−3* mutant; plants were grown on a ½ MS plate in standard conditions. Library preparation and RNA sequencing were performed by OE Biotech Co., Ltd. (Shanghai, China). The paired-end RNA-seq sequencing library was sequenced with the Illumina Novaseq 6000 (2 × 150 bp read length) system. Three biological replicates per sample were analyzed. The clean reads were deposited into the NCBI Sequence Read Archive database (Accession Number: PRJNA820164). *p* value < 0.05 and fold change ≥1.5 or fold change ≤0.67 were set as the thresholds for significantly differential expression. The GO enrichment analysis was based on the biological process functional categories of ShinyGo v0.75 software (http://bioinformatics.Sdstate.edu/go/, date of access 1 March 2022) [32].

### 2.5. True Leaf and Root Growth Inhibition Assays

In the true leaf inhibition assay, more than 100 seeds were sown and grown on a standard ½ MS plate, with or without DNA damage reagent, 15 μg·mL*^−^*^1^ Zeocin (Solarbio, Cas11006-33-0, Beijing, China) or 80 ppm MMS (Sigma, Cas66-27-3, St. Louis, MO, USA). After 12 days, the number of true leaves was observed and counted for seedlings. For the root growth inhibition assay, seeds were sown and grown vertically on ½ MS plates, with or without DNA damage reagent, 3 μg·mL*^−^*^1^ Zeocin or 80 ppm MMS. After 7 days, their root growth was measured using ImageJ software.

### 2.6. Genotoxic Treatments

To investigate the expression of DDR genes after genotoxic treatment, 12*−*d*−*old seedlings grown under the growth conditions were transferred to and soaked in 100 μg·ml*^−^*^1^ Zeocin or 150 ppm MMS for 2 h. Seedlings were then harvested immediately after each stress treatment and subjected to gene expression analysis.

### 2.7. γ-H2AX Assays

For the γ*-*H2AX assays, total proteins were extracted with protein extraction buffer. The supernatant collected after centrifugation was transferred to a new tube, mixed with SDS loading buffer and boiled at 95 °C for 5 min. The protein samples were subjected to 12% SDS-PAGE gel electrophoresis, blotted and immunodetected with rabbit anti*-*γ*-*H2AX antibody (1 μg·mL*^−^*^1^, ab2893, Abcam, Cambridge, UK). *At*GAPDH was used as an internal reference, detected using anti*-*GAPDH antibody (1:5000, 10494*−*1*−*AP, Proteintech Group, Inc Rosement, IL, USA). The band intensities on the immunoblots were determined by analyzing the exposed film with Quantity One professional grayscale analysis software.

### 2.8. Flow Cytometry

Approximately 0.5 cm^2^ of the mature 1st to 2nd true rosette leaf of each plant (18*−*d*−*old) was chopped with a sharp razor blade in 400 µL of ice-cold CyStain UV Precise P nuclear lysis solution (Sysmex, Norderstedt, Germany), for 30*−*60 s, to fully extract the complete cell nuclei. The nuclear suspension was filtered through a CellTrics 30 μm filter (Sysmex) directly into the sample tube, and 1600 µL of staining buffer (DAPI) was added. For the analysis of the nuclei, a Sysmex CyFlow^®^ Ploidy Analyser with UV excitation by a mercury arc lamp was used, and two or more samples were analyzed, with each sample containing at least 5000 nuclei. The data analysis was carried out with FCS Express version 3 software. The endoduplication index (EI) was calculated using the following equation: EI = (0×% 2C) + (1×% 4C) + (2×% 8C) + (3×% 16C) [33].

### 2.9. Leaf Epidermal Cell Examination

The 1st to 2nd true leaf of each 18*−*d*−*old plant of the *rfc4e* mutants and the wild-type Col*−*0 was collected separately in a 15 mL tube and treated with a chloral hydrate:glycerol:water solution (8:1:2) to clear the cells [34]. The epidermal cells on the abaxial leaf surfaces were photographed with a Zeiss Imager M2 microscope, and their leaf size was measured using ImageJ software.

## 3. Results

### 3.1. rfc4e Mutants Were Generated by Site-Directed Mutagenesis

To investigate the functional details of *AtRFC4* in *Arabidopsis*, we ordered a T*−*DNA insertion line, Salk_049715 (*rfc4−1/RFC4*) [12], from the Arabidopsis Biological Resource Center (ABRC). *rfc4−1**/RFC4* carried a T*−*DNA fragment inserted in the second intron (Appendix A) and had a phenotype consistent with the wild type (Col*−*0), but *rfc4−1* was found to be lethal, as indicated by the absence of homozygotes in the progeny of heterozygous lines, meaning that the complete RFC4 protein is indispensable for *Arabidopsis*. According to previous studies, the integrity of the C-terminus of the RFC is crucial for the formation of the RFC complex. Therefore, frameshift mutation of the *RFC* gene will lead to lethality in *Arabidopsis* due to the inability to form a complete RFC complex. For the above reasons, and inspired by our previous study, a point mutant of *RFC3*, another subunit of the RFC complex, was obtained by EMS mutagenesis. The 84th Gly in the conserved Walker A domain of RFC3 was found to be mutated into the acidic amino acid Aspartic [29]. Bioinformatics analysis showed that the Walker A domain is very conserved in RFC subunits [11] (Figure 1A). Therefore, we decided to substitute the 54th Gly in the Walker A domain of *At*RFC4 with Aspartic or Glutamate by site-directed mutation [30], corresponding to the 84th Gly in *At*RFC3 based on the amino acid alignment (Figure 1A), and *rfc4−1/RFC4* heterozygotes were used as recipients of the site-specific mutagenesis plasmid.

Through phenotype analysis of the offspring, six envisioned transformants of *rfc4−1/RFC4^G54E^* (*rfc4e*) were obtained from 24 hygromycin-resistant plants (Figure 1B). To further identify the *RFC4^G54E^* insertion site, *rfc4e−1*, *rfc4e−2* and *rfc4e−3* were sequenced by next-generation sequencing (NGS) and identified by PCR, and all of the insertion sites were found to be located in the untranslated region of the genome (Figure 1C). *rfc4e* mutants show a dwarf phenotype, being smaller in size and lighter than Col*−*0 (Figure 1D,E). RT*-*qPCR analysis showed that *RFC4^G54E^* could be expressed normally in three homozygous mutants, and the expression level in *rfc4e−2* and *rfc4e−3* was more than twice that in *rfc4e−1* (Appendix A). To further verify that the defective phenotype was caused by the mutation of *AtRFC4*, a complementation test was conducted by transforming the full-length genomic fragment of wild-type *AtRFC4* into *rfc4e−3* mutants. Two independent complemented lines were then rescued (Appendix A), indicating that the mutation of *AtRFC4**^G54E^* is responsible for the dwarf phenotype of *rfc4e* mutants, and most likely a partial loss-of-function mutation.

Unexpectedly, the *rfc4−1/RFC4^G54D^* mutant was lethal, as indicated by the absence of homozygous mutants in the progeny (*n* > 200), suggesting that the conserved amino acid G to D substitution in *At*RFC4 is deleterious to the survival of offspring, differing from *At*RFC3.

### 3.2. Mutation of AtRFC4 Causes Earlier Flowering and Seed Abortion

All three *rfc4e* mutants showed similar and pleiotropic developmental phenotypes, including small and narrow serrated leaves, early flowering and partial sterility (Figure 2A). The size of flowers and petals, as well as the length of siliques, of *rfc4e−1, rfc4e−2* and *rfc4e−3* were noticeably smaller than those of the wild type when grown on soil (Appendix A). Under long-day conditions (16 h light/8 h dark), wild-type plants flowered at ~35 ± 2 d after germination, with ~14 ± 2 leaves, while *rfc4e* plants flowered at ~21 ± 2 d after germination, with ~8 ± 1 leaves (Figure 2B). Consistent with the early flowering phenotype, expression of *Flower Locus T* (*FT*) (Figure 2C) and *Apetala 1* (*A**P1*) (Figure 2D) was much higher in *rfc4e* than that in wild-type plants, although the expression of the flowering suppressor gene *Flower Locus C* (*FLC)* was not significantly different between them (Figure 2E).

Furthermore, when the fertility of *rfc4e* plants was analyzed 2 weeks after pollination, *rfc4e* was found to produce fewer seeds than wild-type plants. On average, 29.74, 38.81 and 39.60 seeds per silique were produced in *rfc4e−1, rfc4e−2* and *rfc4e−3*, respectively, while 46.87 seeds per silique were found in the wild type (Figure 2F and Appendix A). In contrast to wild-type plants, siliques from *rfc4e−1, rfc4e−2* and *rfc4e−3* produced 61.21%, 27.69% and 30.35% aborted seeds, respectively (Figure 2G), indicating that *RFC4* mutation causes earlier flowering and seed abortion.

### 3.3. DNA Repair- and Replication-Related Genes Are Highly Upregulated in rfc4e Mutants

To further investigate the effect of the *rfc4e* mutation, we performed an RNA sequencing experiment using the aboveground part from the 12*−*d*−*old Col*−*0 and *rfc4e−3* plants grown on ½ MS plates. Volcano plot analysis revealed that 1307 genes were significantly upregulated and 1114 genes were significantly downregulated (*p* < 0.05, |Log_2_FoldChange| ≥0.58, Figure 3A). Considering all DEGs in *rfc4e−3*, a Gene Ontology (GO) term enrichment analysis based on the biological process category showed an overrepresentation of genes involved in development and primary metabolic pathways, such as the biosynthetic process, cellular biosynthetic process and other cellular responses, most of them related to stress responses (Appendix A), which is consistent with the severe defects in *rfc4e* mutants.

Remarkably, by filtering DEGs based on the 1307 upregulated genes, the top 10 fold enrichment results of the GO term enrichment analysis highlighted their involvement in cell cycle progression, DNA replication progression, the DNA damage response and repair process, as well as the stress response (Figure 3B), and the downregulated genes were mainly related to cellular starvation and stimuli (Appendix A). Moreover, when the enrichment of upregulated genes was analyzed using the Kyoto Encyclopedia of Genes and Genomes (KEGG) [35], only five enriched pathways were identified (Appendix A), including homologous recombination (HR), DNA replication, mismatch repair, base excision repair and nucleotide excision repair. Interestingly, all of these pathways are closely related to DNA repair and replication.

To validate the KEGG data, HR*-*related genes *BRCA1* [36], *RAD51* [37] and *RAD54* [38], DNA damage sensors *PARP1* and *PARP2* [39], *RPA* (*replication protein A*) [40], *PCNA* (replication clamp gene) [41], *KU70/80* (non*-*homologous end joining (NHEJ) repair pathway initiation gene), *Pol*
*λ* (DNA polymerase λ encoding gene) and *ligase 4-XRCC4* (X*-*ray cross*-*complementary protein 4 encoding gene) [42,43] were selected as markers for distinct DNA damage pathways, and their expression was monitored for comparison by reverse transcription quantitative PCR (RT*-*qPCR) between Col*−*0 and the *rfc4e−3* mutant under standard culture conditions. Compared with Col*−*0, DDR-related genes including *BRCA1*, *RAD51*, *RAD54*, *PARP2*, *PARP1*, *RPA1E* and *PCNA1* were all upregulated under standard growth conditions in *rfc4e−3*, but the NHEJ*-*related genes *KU70*, *KU80*, *Pol λ* and *Lig4* were almost the same as the wild type (Figure 3C). As expected, a similar expression pattern was found in the *rfc3−1* mutant, which carried a point mutation in RFC subunit 3, indicating that DDR genes were commonly upregulated by the partial loss of function in the RFC complex. Altogether, these data indicate that a defect in *RFC4* causes a constitutive induction of the DDR and the enhanced transcription of genes involved in the HR rather than the NEHJ pathway.

### 3.4. AtRFC4 Mutation Increases the Frequency of DNA Lesions

The constitutive transcriptional DDR suggests that *rfc4e* mutants are subjected to genome instability. One of the well*-*known immediate responses to the induction of DSBs includes the rapid phosphorylation of large numbers of the histone variant H2AX precisely at the site of DNA damage [44]. Thus, the phosphorylation of histone variant H2AX was tested to check the frequency of DNA lesions. In this context, 12*−*d*−*old Col*−*0 and *rfc4e−1, rfc4e−2* and *rfc4e−3* plants grown on ½ MS plates were detected in immunoblotting experiments with equal amounts of protein extracts using γ*-*H2AX (S139) phosphorylation antibody (Abcam, ab2893), which recognized a 17 kDa protein. Although the antibody could also recognize low levels of γ*-*H2AX protein in extracts from Col*−*0 seedlings (Figure 4A,B), the mutated plants exhibited more than 2.7*−*fold higher accumulation of γ*-*H2AX signals, indicating the presence of more DSBs in *rfc4e−1, rfc4e−2* and *rfc4e−3* plants, reflecting an excessive frequency of DNA lesions even in standard conditions.

### 3.5. rfc4e Mutants Are Supersensitive to DSB-Inducing Genotoxic Agents

Mutation of *AtRFC1* and *AtRAD17* was previously shown to confer upon plants sensitivity to UV irradiation and DNA toxic substance treatment [25,45]. To check whether the resistance to DNA damage repair was also compromised due to the accumulation of DNA lesions in *rfc4e* plants, *rfc4e−1*, *rfc4e−2*, *rfc4e−3* and Col*−*0 were planted on media containing different concentrations of Zeocin or methyl methanesulfonate (MMS) (Figure 5A).

Radiomimetic reagent Zeocin is a bleomycin family antibiotic causing DNA damage by cleaving both strands of the DNA molecule-induced DSBs [24], while MMS causes base mispairing and replication fork stalling [46]. These types of DNA damage were found to be repaired mainly by homologous recombination. As shown in Figure 5B, when treated with 80 ppm MMS or 15 μg·mL^−1^ Zeocin, *rfc4e−1*, *rfc4e−2* and *rfc4e−3* mutants were severely retarded, with lower numbers of true leaves than Col*−*0 plants (Figure 5B).

In a root elongation assay, *rfc4e* mutants showed shorter lateral roots (on average, 14.05 mm, 13.30 mm and 15.94 mm in *rfc4e−1, rfc4e−2* and *rfc4e−3*, respectively) than the wild type (34.84 mm) (Figure 5C,D). When vertically grown and treated with 3 μg·mL^−1^ Zeocin on ½ MS for 7 days, the inhibition ratio of *rfc4e−1* (79.76%), *rfc4e−2* (82.34%) and *rfc4e−3* (83.52%) was slightly higher than that of Col*−*0 (74.76%) (Figure 5E). Notably, the lateral roots of *rfc4e* mutants were more sensitive than Col*−*0 to 80 ppm MMS treatment. The inhibition rate of *rfc4e* roots reached approximately 78*−*81%, whereas the root inhibition rate of Col*−*0 was 68.67% (Figure 5F). Together, these data demonstrate that *rfc4e* mutants are more sensitive to DDR regents, MMS and Zeocin, due to the accumulation of DNA lesions.

Moreover, consistent with the transcriptome data, a number of DDR pathway genes, including *NAC103*, *NAC053*, *RAD51*, *PARP1*, *PARP2*, *BRCA1*, *GR1*, *RPA1E* and *RAD17*, were highly induced in *rfc4e−1*, *rfc4e−2* and *rfc4e−3* plants under normal conditions (Figure 5G). When treated with 100 μg·mL^−1^ Zeocin for 2 h, all these genes were also induced both in Col*−*0 and *rfc4e*, but the relative induction was lower in *rfc4e* than in the Col*−*0 control (Figure 5H). These results suggest that DDR pathway genes are constitutively induced in *rfc4e* mutants due to the impaired repair ability, leading to hypersensitivity to DSB-induced genotoxic stress, indicative of the DDR-regulatory function of *At*RFC4.

### 3.6. AtRFC4 Mutation Delays Cell Cycle and Promotes Endoduplication

Since DDR activates cell cycle abnormality and mutation of RFC3 leads to smaller numbers of cells in true leaves, the replication-related phenotype was further examined in *rfc4e* mutants. As shown in Figure 6A, all *rfc4e* mutant alleles exhibited dwarf status, with smaller and narrower leaves compared with wild-type plants. The leaf blades of *rfc4e* mutants were less than half the size of wild-type blades, approximately 37.64% (*rfc4e−1*), 43.72% (*rfc4e−2*) and 44.44% (*rfc4e−3*) for the second true leaves, respectively (Figure 6B). To determine whether the smaller leaves of the *rfc4e* mutants were caused by the smaller number or size of cells, the epidermal cells of the first to second true leaves were examined under a microscope. As expected, the epidermal cells on the abaxial surfaces of the first to second true leaves in *rfc4e* mutants were much larger than those of the corresponding wild-type true leaves (Figure 6C,D). As a result, with larger cells but a smaller leaf area, the epidermal cell number on the abaxial surface of the first to second true leaves was only 17.21%, 25.14% and 27.37% in *rfc4e−1*, *rfc4e−2* and *rfc4e−3*, respectively, of the corresponding cell number in Col*−*0 plants.

As replication-related defects often result in endoduplication, flow cytometry analysis of the first to second true leaves of wild-type plants and *rfc4e* mutants was performed (Figure 7A). In *rfc4e* mutants, the proportion of 16C nuclei was higher than in the wild type, while the proportion of 2C nuclei decreased (Figure 7B). In addition, the 8C/2C, 16C/2C ratio (Figure 7C) and endoduplication index (Figure 7D) were significantly increased in *rfc4e* mutants. Moreover, some of the cell-cycle-related genes, including *AtCYCB1;1*, *AtWEE1*, *AtSMR4*, *AtSMR5* and *AtSMR7*, were also upregulated in the *rfc4e* mutants compared with wild-type seedlings (Figure 7E). These results indicated that a large number of cells were arrested in the G2 phase rather than entering into the mitotic phase, which led to increased endoduplication in *rfc4e* mutants.

## 4. Discussion

In the past few decades, many studies have provided details of the genes involved in the DNA replication and DNA damage repair process. However, due to the lethality of mutation, the four smaller subunits of the RFC complex have rarely been studied in higher plants. In this study, we generated viable mutants of *RFC4* with a single amino acid substitution by means of site-directed mutagenesis. Compared to Col*−*0, *rfc4e* mutants were smaller, flowered earlier, and produced a large number of aborted seeds, showing a DNA replication-defective phenotype (Figure 1 and Figure 2), similar to *rfc1−1* [25] and *rfc3−1* [29]. Interestingly, *rfc4e−1* showed a higher abortion rate than *rfc4e−2* and *rfc4e−3*—this may be due to the relatively lower expression of *RFC4^G54E^* in *rfc4e−1* (Appendix A). Consistent with this, cell division in embryos and endosperm was shown to be inhibited in *rfc4−1* heterozygotes, leading to the embryo lethality of the *rfc4−1* homozygotes in a former study [12]. These results indicate that the single amino acid substitution in the conserved Walker A domain seriously affects the normal function of RFC4 and is not conducive to the adaptation of plants to the natural environment, but the abortion defect can be alleviated by an increase in *RFC^G54E^* expression.

Transcriptome data analysis showed that the expression of DNA damage repair-related genes strongly increased in the *rfc4e−3* mutant, and immunoblotting experiments showed that there were more γ-H2AX signals in *rfc4e−1, rfc4e−2* and *rfc4e−3* mutants, indicating that *AtRFC4* mutation leads to an impaired ability to perform DSB repair. In plants, there are two main modules of DSB repair, HR and NHEJ. GO and KEGG aggregation analysis revealed that HR-related genes were highly expressed in *rfc4e* mutants. Among these genes, *AtRAD51* is an essential recombinase for mitotic and meiotic HR repair [37], while RAD54 [38] is an essential cofactor that stimulates RAD51 activity, which has a significant effect on DNA damage repair in mitotic cells. In addition, *AtBRCA1* is also required for efficient DSB repair by homologous recombination in somatic cells [36]. The significantly increased expression levels of these genes in *rfc4e* mutants suggest that the HR repair pathway might be activated. However, there was no significant difference in the expression levels of NHEJ-related genes such as KU70/80, indicating that HR, rather than NHEJ, might be the main DSB repair mechanism in *rfc4e* mutants. Consequently, *rfc4e* mutants are supersensitive to DNA-damaging agents MMS and Zeocin, and they constitutively express higher levels of DDR-related genes, such as *GR1*, *BRCA1*, *RAD51*, *PARP1, PARP2* and *RPA1E*. Mutations in core replication machinery proteins (such as Pol δ [47]) and replication proteins RFC1 [25] and RAD17 [45] are known to increase somatic homologous recombination (SHR), and they are sensitive to DNA-damaging agents. Moreover, the expression of DDR genes in *rfc3−1* mutants shows a similar expression pattern to that observed in *rfc4e* mutants, further suggesting that the mutation of these core DNA replication factors compromises the capacity for DSB repair during DNA replication and DNA damage in *rfc4e* and *rfc3−1* mutants.

When DNA damage occurs, plant cells have the choice of either delaying cell division to repair the damage or inducing cell death. A likely benefit of the first choice is that entering into endoduplication prevents DNA-damaged cells from proliferating and also from dying. In *rfc4e* mutants, we observed lower numbers of epidermal cells with increased DNA ploidy. Several genes that have been identified as critical for the inward replication of cyclins and cyclin-dependent kinase (CDK) complexes are all upregulated in *rfc4e* mutants. The *Arabidopsis* mitotic cyclin *CYCB1;1* is expressed during the G2/M transition, and its upregulation suggests a G2 arrest [48]. Members of the SIAMESE/SIAMESE-RELATED (SIM/SMR) class of cyclin-dependent kinase inhibitors, including SMR4, SMR5 and SMR7, activate cell cycle checkpoints in response to DNA damage, thereby inhibiting mitosis and promoting endoduplication [49]. The high expression of cell cycle inhibitory protein kinase WEE1 can promote nuclear replication, and its expression in tomato fruit tissue is positively correlated with DNA fold [50]. In *Drosophila,* the mitotic cells of *I(3)Rfc4^e18^* and *I(3)Rfc4^e20^* lines exhibited prematurely condensed chromosomes or chromosomal bridges/breaks [18]. Meanwhile, mutations of *Scrfc2*, *Scrfc5*, *Sprfc2* and *Sprfc3* caused replication defects and chromosomal abnormalities when entering the mitotic stage [14,16,17,51,52]. Based on this evidence, it is plausible to postulate that aberrant mitosis takes place in *rfc4e* mutants. Combined with the increased DSB, embryonic abortion, high expression of DDR genes and endoduplication in *rfc4e* mutants, our data suggest that *AtRFC4* plays an important role in maintaining normal DNA replication and damage repair, and a single amino acid substitution will lead to genomic instability, an impaired repair ability and inward replication transformation of the cell cycle, which ultimately compromises cell proliferation and resistance to genotoxic stresses.

Although there are many methods to study gene function, there are still some genes, which, due to their importance and irreplaceability, are unable to obtain corresponding mutants using EMS mutagenesis or T-DNA insertion methods. The acquisition of RFC4 single amino acid substitution mutants not only enriches the biological function of RFC4 in plants, but also provides a means to study similar housekeeping genes. Above all, the acquisition of mutants is conducive to studying the comprehensive effects of these genes in plants.

## Figures and Tables

**Figure 1 genes-13-01037-f001:**
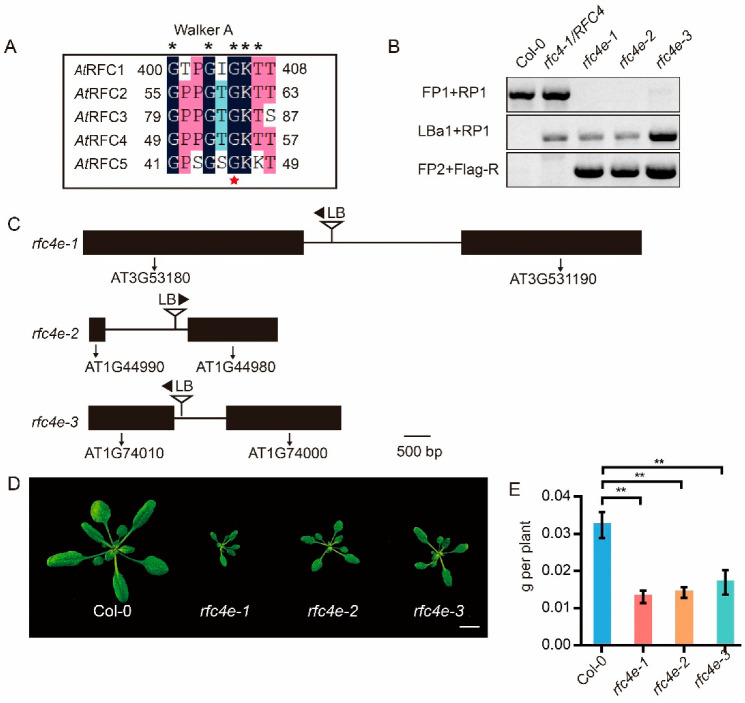
*rfc4e* mutants were generated by site-directed mutagenesis. (**A**) Sequence similarities in the Walker A domain among *At*RFC1/2/3/4/5 subunits. Conserved amino acids in the Walker A domain are indicated with asterisks, and the mutated Gly is indicated with a red star. (**B**) Genotypic analysis of the wild type (Col*−*0), *rfc4−1*/*RFC4*, *rfc4e−1, rfc4e−2, rfc4e−3*. (**C**) Schematic diagrams of T*−*DNA insertion position of *rfc4e−1, rfc4e−2* and *rfc4e−3*, respectively. Black boxes and lines indicate genes and untranslated regions flanked the insertion sites. (**D**,**E**) Phenotype and fresh weight of Col*−*0, *rfc4e−1, rfc4e−2* and *rfc4e−3* seedlings at 18 days after germination. Bar = 1 cm. Error bars indicate ± SD (** *p* < 0.01).

**Figure 2 genes-13-01037-f002:**
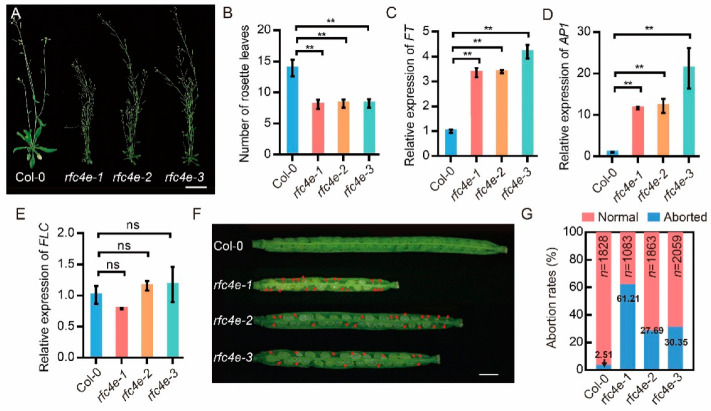
*AtRFC4* mutation causes earlier flowering and seed abortion. (**A**) Morphological phenotype of Col*−*0, *rfc4e−1, rfc4e−2* and *rfc4e−3* at 50 days after germination. Bar = 2 cm. (**B**) The number of rosette leaves when plant blooms, *n* > 50. (**C***−***E**) Relative expression of *FT* (**C**), *AP1* (**D**) and *FLC* (**E**) in *rfc4e−1*, *rfc4e−2, rfc4e−3* and Col*−*0. Total RNA extracted from 12*−*d*−*old seedlings was reverse-transcribed. The expression in Col*−*0 was used as a standard for normalization control. Data are means of triplicate technical repetitions from one of three biologically independent experiments. Error bars indicate ± SD. (**F**) Ovule phenotypes in siliques of Col*−*0, *rfc4e−1*, *rfc4e−2* and *rfc4e−3*. More than 30 siliques were counted, and the aborted ovules are labeled by red arrows. Bar = 1 cm. (**G**) Seed abortion ratios of Col*−*0, *rfc4e−1, rfc4e−2* and *rfc4e−3* plants. The total number of seeds counted is listed at the top of the histogram bars and the seed abortion ratio at the bottom. Pale seeds represented delayed embryos and/or aborted seeds. Error bars indicate ± SD, *n* > 1000 (** *p* < 0.01), ns, no significance.

**Figure 3 genes-13-01037-f003:**
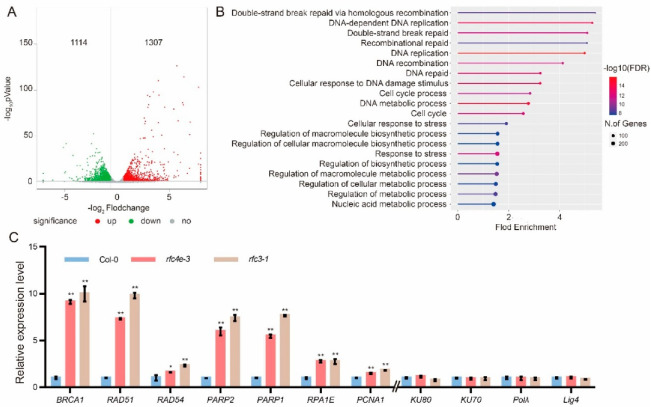
The whole transcriptome of the *rfc4e−3* mutant reveals overexpression of numerous DDR genes. (**A**) Volcano plots showing the differentially expressed genes in *rfc4e−3* compared with wild-type plants (*p* < 0.05, |Log2FoldChange| ≥ 0.58). The red dots represent significantly upregulated genes, and the green dots represent significantly downregulated genes. (**B**) Gene Ontology (GO) enrichment analysis of the 1307 upregulated genes. The size of each dot represents the gene count, and the color of each dot represents the enrichment fold. (**C**) Relative expression levels of gene transcripts from 12*−*d*−*old in vitro-grown plants of the indicated genotype as determined by RT-qPCR. The expression level in Col*−*0 was used as a standard for normalization control. Data are means of triplicate technical repetitions from one of three biologically independent experiments. Error bars indicate ± SD (* *p* < 0.05; ** *p* < 0.01).

**Figure 4 genes-13-01037-f004:**
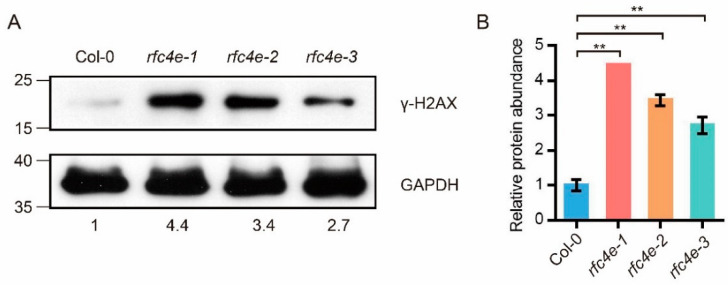
Detection of histone H2AX phosphorylation in wild-type and *rfc4e* plants. (**A**) The 12*−*d*−*old Col*−*0, *rfc4e−1*, *rfc4e−2* and *rfc4e−3* plants were used to extract total protein. Immunodetection of γ*-*H2AX was carried out using rabbit anti*-*γ*-*H2AX polyclonal antibody in total protein extracts. The upper panels in Figure 4A represent Western blots from standard*-*condition seedlings, while the lower panels indicate expression levels of GAPDH protein; anti*-*GAPDH antibody was detected and used as a loading control. Molecular mass markers in kilodaltons are indicated on the left. (**B**) The relative γ*-*H2AX band intensity normalized to loading control, relative to Col*−*0. The experiment was repeated two times with the same tendency. Error bars indicate ±SD (** *p* < 0.01).

**Figure 5 genes-13-01037-f005:**
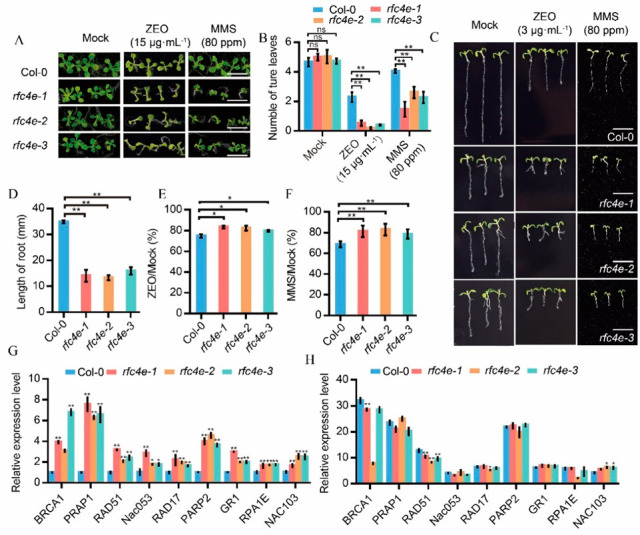
*The rfc4e* mutants were sensitive to DNA damage agents and constitutively expressed DNA repair-related genes. (**A**) The 12*−*d*−*old seedling of *rfc4e−1*, *rfc4e−2, rfc4e−3* and Col*−*0 grown on ½ MS medium containing 80 ppm MMS or 15 μg·mL*^−^*^1^ Zeocin. Mock, treatment without MMS and Zeocin used as control. Bar = 1 cm. (**B**) Average numbers of true leaves of the indicated genotypes in (**A**). Error bars indicate ± SD, *n* = 39. (**C**) Roots of *rfc4e−1*, *rfc4e−2, rfc4e−3* and Col*−*0 vertically grown on ½ MS medium containing 80 ppm MMS or 3 μg·mL*^−^*^1^ Zeocin for 7 days. Mock, treatment without MMS and Zeocin used as control. Bar = 1 cm. (**D**) Root length of the indicated genotypes in (**C**), *n* > 20. (**E**,**F**) Relative root growth rate of Col*−*0 and *rfc4e* mutants grown on medium containing 15 μg·mL*^−^*^1^ Zeocin (**E**), or 80 ppm MMS (**F**). Mock, treatment without MMS and Zeocin used as control. (**G**,**H**) DDR gene expression levels in Col*−*0 and *rfc4e* mutants treated without (**G**) and with (**H**) Zeocin. (**G**) Relative expression of genes in 12*−*d*−*old plants of the indicated genotypes under standard conditions, as determined by RT*-*qPCR. Data were normalized to the value of Col*−*0. (**H**) Relative expression of genes in 12*−*d*−*old plants of the indicated genotypes after 2 h of 100 μg·mL*^−^*^1^ Zeocin treatment, as determined by RT-qPCR. Data were normalized to the value of the same genotype under standard conditions. The experiment was repeated twice with similar results. Error bars indicate ± SD (* *p* < 0.05; ** *p* < 0.01), ns represents no significance.

**Figure 6 genes-13-01037-f006:**
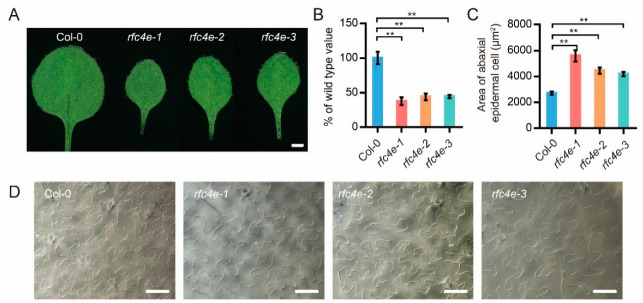
The *AtRFC4* mutation leads to smaller leaves and enlarged lower epidermal cells. (**A**) The second true leaf of the wild type, *rfc4e−1*, *rfc4e−2* and *rfc4e−3*. Bar = 1 mm. (**B**) The average area of first to second true leaves of 18*−*d*−*old seedlings of the indicated genotypes; data are compared to the Col*−*0 value normalized at 100%. *n* ≥ 10. (**C**,**D**) The abaxial epidermal cells (**D**) and their average areas (**C**) of first to second true leaves of 18−d−old seedlings of the indicated genotypes. Bar = 50 μm. Error bars indicate ± SD (** *p* < 0.01).

**Figure 7 genes-13-01037-f007:**
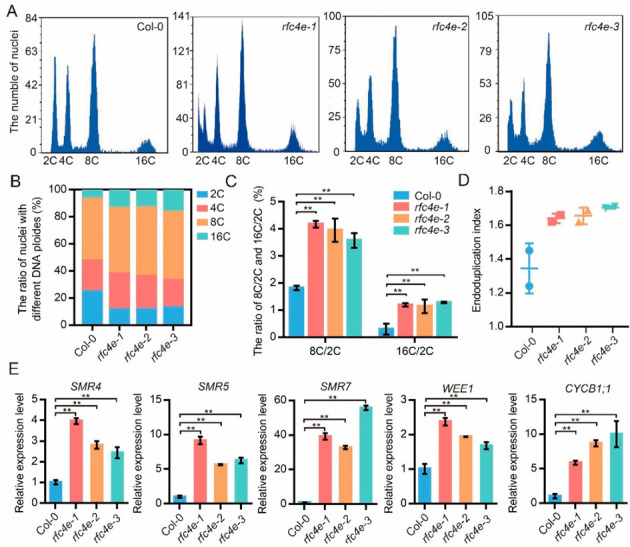
The *AtRFC4* mutation delays the cell cycle and promotes endoduplication. (**A**) Polyploidy in first to second true leaves of 18*−*d*−*old seedlings in Col*−*0, *rfc4e−1*, *rfc4e−2* and *rfc4e−3* plants, respectively. (**B**) The percentage of nuclei with different DNA ploidies of the indicated genotypes. (**C**) The ratio of 8C/2C and 16C/2C of the indicated genotypes. Error bars indicate ± SD (** *p* < 0.01). (**D**) The endoduplication index of the indicated genotypes. (**E**) Relative expression levels of cell cycle genes in 12−d−old seedlings of Col*−*0, *rfc4e−1*, *rfc4e−2* and *rfc4e−3* plants under standard conditions. Data are compared to the Col*−*0 value normalized at 1. The experiment was repeated twice with similar results. Error bars indicate ± SD (** *p* < 0.01).

## Data Availability

RNA sequencing data of *rfc4e-3* were deposited in the NCBI Sequence Read Archive database (Accession Number: PRJNA820164).

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
