# Peer review of "A Single Amino Acid Substitution in RFC4 Leads to Endoduplication and Compromised Resistance to DNA Damage in Arabidopsis thaliana"

_genes, 2022, doi:10.3390/genes13061037_

Round 1

Reviewer 1 Report

This study tried to create a site-directed mutation of one aa substitution in a conserved gene RFC4 in Arabidopsis. Although the experiments are well designed and the data are solid, the real scientific question is not striking. Additionally, the author should discuss the potential cause of the phenotypes of the mutants of the three different insertions (include comparison of the predicted protein structures). The expression of the RFC4 G54E is lower in rfc4e-1 than in the other two, however, the phenotypic effect is more serious in rfc4e-1 than in the other. That means the higher expression of the mutant genes means better fitness. The question is whether the site-directed gene is beneficial or detrimental in terms of fitness in Arabidopsis organisms. Even if the mutant is not selected by natural conditions, the author should compare the fitness effect of ATRFC4 and RFC4 G54E in terms of the Arabidopsis organism. This result also indicates an unexpected scenario: even if the amino acid substitution occurred in the most conserved domain of a housekeeping gene, the plant individual could still survive in lab environments. This should also be discussed in the paper.

minor revisions:

Line13: “AAA+ ATPase”, please compile the full name.

Line16: “rfc4e (rfc4-1/rfc4G54E)”, this is confusing, I feel that rfc4G54E should be RFC4G54E.

Line17: “three mutant alleles”, the three mutants have the same aa substitution but different insertion locus, I could get such information only after I read the whole paper. please clarify here to show a clear description of the basic info.

Line41: “due to damage to the C-terminus”, Just to confirm, T-DNA could cause various defects on DNA which may even cause absent expression, for the T-DNA insertion mutants of AtRFC2-5, if the T-DNAs cause no expression, then how do you know the lethal is caused by the damage to the C-terminus?

Line145: “2.7γ-. H2AX Assays”, punctuation errors.

Line176: “rfc4-1 was found to be lethal”, heterozygous Salk_049715 (rfc4-1/RFC4) is dwarf or normal as WT? Since the author described the dwarf phenotype of the site-directed mutant, however, they didn’t mention the phenotype of heterozygous Salk_049715 (rfc4-1/RFC4).

Line181: “we decided to mutate the 54th Gly in Walker A domain of AtRFC4 into Aspartic or Glutamate”, this means the 54th Gly in AtRFC4 is corresponding to the 84thGly in AtRFC4 based on the aa alignment. Please clarify here or cite Figure 1A.

Line184: “rfc4-1/rfc4G54E (rfc4e)”, It seems that the author didn’t mention whether both loci are hete- or homo-.

Line185: “24 hygromycin resistant plants” should produce 24 potential positive plants, why the other 18 are not positive even they carry the hygromycin resistant genes?

Line187: “When these three mutants were sequenced”, why the author choose these three but not all six?

Line177,178,179, 194: “which is different from AtRFC3”, The author tried to compare the phenotypes between the two mutants, however, they didn’t really describe the discrepancy. Additionally, the AtRFC3 mutant is an EMS while AtRFC4 is a site-directed mutant with a T-DNA heterozygous mutant background (the author didn’t mention whether it is a hete- or homo- background) of the original AtRFC4 gene, is this comparison reasonable?

Line291: Figure4B. error bar is missing.

Line300: Format mistake.

Figure S1: Please note the first exon should include the 5UTR region, FigureS1A should be revised.

Figure S2: (B) should be (D)

Author Response

We genuinely thank you for your constructive and helpful suggestions and comments, which help us to improve the quality and thoroughness of our study. The manuscript texts have since been revised substantially. Some minor mistakes and typos were also corrected during revision. Our response in red font can be found beneath each original reviewer’s comment below.

Reviewer 2 Report

In this manuscript, a substitution mutant in RFC4 gene was generated and analyzed phenotypically. The author found that mutant plants were dwarfed, show early flowering and partial sterile phenotype. Mutants were hypersensitive to double-strand break (DSB) inducing agents and show upregulation of genes related to DNA repair and replication, as well as delayed cell cycle progression and endoduplication. The authors conclude that RFC4 might play an important role in DNA repair through homologous recombination.

Overall, the manuscript is sound and presents enough data to support the conclusions.

Some comments:

Because homozygous mutants for the various RFC subunits are lethal, reverse genetics strategies are not viable, thus partial loss-of-function mutants have been employed. A previous example on RFC3 was cited and employed to generate the Gly to Glu mutation. However, the authors do not discuss whether it is a partial loss-of-function allele.

The first part of the Results section, where the generation of the mutant is described, is quite confusing and should be rewritten. For example, in the last paragraph of page 4: … , “and the relative expression level of RFC4 or RFC4 G54E was shown in Figure S1B”. However, there is no further explanation on the presented in Figure S1B, as not all three alleles mutants have the same behavior.

The manuscript should be thoroughly revised for grammar, spelling and syntax. For example, in legend of figure 1, it is written “wight”, instead of “weight”.

The Discussion section is insufficient and does not clearly presents all the implications of the study. The authors refer to SOG1 as an important regulator in DNA Damage Response, so one question would be whether RFC4, or any other RFC subunits are targets of SOG1?

Author Response

(The authors gave the same response as above.)
